# Construction of an intelligent screening model for allergic rhinitis based on routine blood tests

**Change Fan**[ID]1*, **Yanan Wang**2, **Xin Tong**1, **Shiyu Wu**2, **Caiyan An**1, **Huijiao Cai**1, **Junjing Zhang**1, **Biao Song**2, **Ruihuan Zhang**2

**1** Hohhot First Hospital Allergy Center, Hohhot, China, **2** Inner Mongolia Zhihui Big Data Research Institute, Hohhot, China

* 384370060@qq.com

## Abstract

The incidence of allergic rhinitis (AR) has been increasing annually, severely impacting patients' quality of life and increasing socioeconomic burdens. The limitations of current diagnostic methods have made the development of efficient, low-cost early screening tools urgent. Based on routine blood test data, this study employed an ensemble hard voting strategy, a comprehensive filtering strategy, an embedding strategy, and a packing strategy to select 16 highly correlated features with a frequency of at least two occurrences as model inputs. Subsequently, the top three machine learning algorithms (K-nearest neighbor, logistic regression, random forest, decision tree, and support vector machine) were selected based on the area under the curve (AUC) metric as the base classifiers. An intelligent early screening model for AR was constructed using an ensemble soft voting strategy. This model demonstrated superior performance, achieving an AUC of 0.862, significantly outperforming any single algorithm. Furthermore, the external validation accuracy was 73.91%. These results demonstrate that combining an ensemble voting strategy with machine learning methods can effectively construct an early screening model for AR based on routine blood test parameters without adding additional burden to patients, providing a new approach to improving diagnosis and treatment in primary care settings.

## Introduction

Allergic rhinitis (AR) is a global disease caused by complex interactions between genetic and environmental factors and affects 400 million people worldwide. Environmental exposure, climate change and lifestyle are all risk factors for AR. For example, pollen in the air increases the prevalence of AR [1,2]. In recent years, the incidence of this disease has risen sharply [3]. In Denmark, the prevalence of AR among the adult population has gradually increased from 19% to 32% over the past three decades [4]. Similarly, the standardized prevalence of AR among adults in China has increased by 6.5% over the past 6 years [5]. The self-reported

**Data availability statement:** All relevant data are within the paper and its Supporting Information files.

**Funding:** The author(s) received no specific funding for this work.

**Competing interests:** The authors have declared that no competing interests exist.

prevalence of pollen-induced AR has reached an extremely high value of 32.4% in the grassland areas of northern China [6]. The high incidence of AR not only seriously affects the quality of life of patients but also results in a heavy economic burden. Currently, screening for AR mainly relies on a comprehensive evaluation of clinical manifestations, physical signs and allergen detection. However, these traditional methods have certain limitations in large-scale screening. For example, questionnaires rely on the subjective responses of the examinees, which may lead to inaccurate information; skin prick tests (SPTs) [7] are strongly affected by drugs and skin conditions and have the potential for severe allergic reactions; and serum allergen-specific IgE (sIgE) tests [8,9] have low sensitivity and high costs, increasing the economic burden on patients. Therefore, there is an urgent need for a new method for AR screening that is objective, accurate, and highly applicable.

Routine blood tests are routine tests with low prices and are suitable for large-scale population screening. Studies have shown that patients with allergic rhinitis have certain manifestations according to routine blood test indicators, especially an increase in eosinophils, which not only reflects the allergic state of the body but also reflects the severity of AR. Therefore, routine blood test data, as an objective examination method suitable for large-scale AR screening, can effectively solve the subjective problem in questionnaire descriptions.

In recent years, with the rapid development of science and technology, artificial intelligence (AI) has demonstrated a profound influence in various fields [10,11], especially in data processing and diagnostic assistance, and its potential has surpassed that of traditional methods. Moreover, China's medical informationization level is in a stage of rapid development. The volumes of various types of clinical test data are growing rapidly, indicating the necessity of medical reform based on big data technology. However, the existing single machine learning methods have problems such as low stability and limited expression ability when processing complex data, resulting in low accuracy of prediction and decision-making. The integrated voting method [12,13] combines the prediction results of multiple single basic models to obtain more stable prediction results, thereby improving the accuracy and robustness of the model and reducing model bias and variance. Therefore, based on routine blood test data, this paper adopts an integrated hard voting method, comprehensive filtering method, embedding method and wrapping method to screen test indicators strongly correlated with AR. An integrated soft voting method is adopted, combining the top three algorithms among K-nearest neighbors (KNN), logistic regression (LR), random forest (RF), decision tree (DT) and support vector machine (SVM) methods to construct an accurate convenient and universal AR screening model, expand the theoretical method system for objective evaluation of AR, compensate for the defects of existing AR screening methods, reduce the disease and economic burden of AR on the population, and provide a scientific basis for formulating more effective prevention measures and control strategies.

## Materials and methods

### Data source

This study included AR patients who visited the outpatient department of the First Hospital of Hohhot from March 21, 2023 to September 3, 2023. Specifically, the data from 26 routine tests (Table 1) and the AR diagnosis results of patients aged 18–70 years were included. The study was approved by the Ethics Committee of Hohhot First Hospital(Approval No: [IRB2022018]). All procedures performed in studies involving human participants were in accordance with the ethical standards of the institutional and/or national research committee and with the 1964 Helsinki declaration and its later amendments or comparable ethical standards. Informed consent was obtained from all individual participants included in the study. Prior to data collection, the purpose of the study, potential risks and benefits, and the confidentiality of their information were explained in detail to all participants. To ensure privacy and confidentiality, all personally identifiable information (PII) of participants was removed during data preprocessing. This included, but was not limited to, names, identification numbers, and contact information. Each patient was assigned a unique, non-identifiable code for data analysis. The dataset used for model development and validation was fully anonymized. The subjects were divided into an AR group and a non-AR group. The inclusion criteria for the AR group were as follows: (1) the main symptoms of AR

**Table 1. Statistical results of blood routine test data based on SPSS.**

| Characteristic | Full name | Negative and positive statistical results | Positive result | Negative result | P-value |
|---|---|---|---|---|---|
| sex | | 0(61.69%) | 0(75.61%) | 0(68.34%) | <0.001 |
| age | | 43.38±11.74 | 40.65±10.17 | 46.36±12.59 | <0.001 |
| RBC | Red blood cell count | 4.8±0.46 | 4.8±0.47 | 4.81±0.45 | 0.687 |
| MCV | Mean corpuscular volume | 92.02±5.46 | 91.84±4.82 | 92.21±6.09 | 0.224 |
| PDW | Platelet distribution width | 13.51±1.8 | 13.3±1.71 | 13.73±1.87 | <0.001 |
| WBC | White blood cell count | 6.34±1.6 | 6.37±1.57 | 6.32±1.64 | 0.528 |
| NEUT% | Neutrophil ratio | 56.69±7.79 | 56.77±7.73 | 56.61±7.87 | 0.706 |
| LYMPH% | Lymphocyte ratio | 33.99±7.42 | 33.35±7.22 | 34.69±7.58 | 0.001 |
| EO% | Eosinophil ratio | 1.7(1.,3.) | 2.4(1.4,3.9) | 1.3(0.8,2.) | <0.001 |
| BASO% | Basophil ratio | 0.5(0.3,0.6) | 0.5(0.3,0.6) | 0.5(0.3,0.6) | 0.459 |
| NEUT# | Neutrophil absolute value | 3.49(2.8,4.26) | 3.66±1.21 | 3.49(2.76,4.26) | 0.385 |
| BASO# | Absolute basophil count | 0.03(0.02,0.04) | 0.03(0.02,0.04) | 0.03(0.02,0.04) | 0.408 |
| HGB | Hemoglobin | 143.23±17.02 | 143.03±16.46 | 143.44±17.62 | 0.671 |
| HCT | Hematocrit | 0.42±0.11 | 0.44±0.04 | 0.43(0.4,0.47) | <0.001 |
| MCH | Mean corpuscular hemoglobin content | 29.81±2.15 | 29.81±1.9 | 29.82±2.39 | 0.912 |
| MCHC | Mean corpuscular hemoglobin concentration | 323.88±11.42 | 324.49±10.49 | 323.22±12.33 | 0.047 |
| RDW-CV | Red blood cell distribution width | 13.37±1.48 | 13.19±1.41 | 13.57±1.53 | <0.001 |
| PLT | Platelet count | 256.9±60.81 | 254.34±55.5 | 259.69±66.06 | 0.117 |
| MPV | Mean platelet volume | 11.13±1.21 | 11.11±1.23 | 11.15±1.2 | 0.605 |
| PCT | Platelet count | 0.28±0.06 | 0.28±0.06 | 0.29±0.07 | 0.099 |
| MONO# | Absolute value of monocytes | 0.4±0.13 | 0.4±0.13 | 0.41±0.14 | 0.634 |
| MONO% | Monocyte ratio | 6.42±1.5 | 6.38±1.47 | 6.48±1.53 | 0.219 |
| EO# | Absolute eosinophil count | 0.11(0.06,0.18) | 0.15(0.08,0.23) | 0.08(0.05,0.13) | <0.001 |
| IG# | Immature granulocyte count | 0.01(0.01,0.01) | 0.01(0.01,0.01) | 0.01(0.01,0.01) | 0.484 |
| RDW-SD | Red blood cell distribution width – SD | 44.16±3.57 | 43.22±3.48 | 45.18±3.39 | <0.001 |
| IG% | The percentage of immature granulocytes | 0.1(0.1,0.2) | 0.1(0.1,0.2) | 0.1(0.1,0.2) | 0.843 |
| P-LCR | Large platelet ratio | 30.78±7.85 | 30.37±7.56 | 31.23±8.14 | 0.049 |
| LYMPH# | Absolute lymphocyte count | 2.12±0.57 | 2.08±0.54 | 2.15±0.6 | 0.034 |

were nasal congestion, nasal itching, runny nose, or sneezing, including two or more symptoms; (2) the allergen skin prick test was positive at 3–5 mm, and the allergens included Artemisia grandis, Artemisia annua, Humulus, Chenopodium album, corn, sunflower, poplar, willow, elm, cypress, birch, house dust mite, dust mite, Alternaria alternata, cat hair, dog hair, ragweed, and Artemisia sphagnum. The inclusion criteria for the non-AR group were as follows: (1) the history of AR was denied; and (2) the allergic skin prick test was negative. According to the above inclusion criteria, those whose clinical data were incomplete and affected the judgment were excluded. Eliminate sample data with missing values or outliers. Finally, routine examination data and AR diagnosis results were collected for 1295 subjects, including 676 AR patients and 619 non-AR patients.

## External validation set

This study used a single-center prospectively recruited external validation cohort. The data came from outpatients in the Allergy Department of Hohhot First Hospital from March 21 to September 3, 2023. After screening and meeting the inclusion criteria, a total of 184 patients aged 19–70 years (49 males and 135 females) were included; according to clinical gold standards (such as allergen sIgE testing and typical symptoms and signs), they were divided into 73 allergic rhinitis (AR) patients and 111 non-AR control groups. The comprehensive model was employed to predict the probability of AR for each patient. A probability score ≥ 0.5 was regarded as positive, while a score < 0.5 was classified as negative. The model's predictions were subsequently compared against the actual clinical diagnoses to evaluate the screening performance of the integrated voting method.

## Statistical analysis

Before model construction, to optimize the effect of feature screening, this study used SPSS software for statistical analysis based on 28-dimensional features (26 routine blood test indicators and age and gender). The statistical methods selected were as follows: (1) Qualitative variables were tested using the chi-square test and expressed as categories (percentages); (2) Quantitative variables were first tested for normality. If the variables were normal (Shapiro-Wilk test $p > 0.05$) or approximately normal distribution (standard deviation < mean/3), the independent sample t-test (mean ± standard deviation) was used, and significance was determined by the Levene test (Student's t-test was used when $p > 0.05$, otherwise the Welch's t-test was used); (3) Non-normal or approximately normal distribution variables were tested using the Mann-Whitney U test and expressed as median (upper quartile, lower quartile). Specific feature abbreviations, definitions, and statistical results are shown in Table 1.

### Routine blood feature screening based on integrated hard voting

In terms of feature selection, filtering, embedding and wrapping methods each have their own advantages: filtering uses statistical indicators (such as chi-square test and mutual information) to perform preliminary feature screening independently of the model, avoiding model assumption bias and being suitable for high-dimensional data preprocessing [14]; embedding embeds feature selection into the model training process (such as LASSO or feature importance of tree models), automatically eliminates redundant features through regularization or integration mechanisms, and is particularly suitable for processing nonlinear relationship features [15,16]; wrapping is guided by model performance (such as recursive feature elimination, RFE), and captures feature interaction effects through iterative search, and performs outstandingly in scenarios such as medical diagnosis that require fine optimization of feature combinations [17]. Based on the above advantages, this study selected these three methods for feature screening experiments.

This study proposes a robust feature selection method based on an integrated voting strategy, which constructs an optimal feature subset by integrating the advantages of filtering, embedding, and wrapping. The specific implementation is divided into two stages: the first stage adopts multi-strategy parallel screening, in which the filtering method selects the

top 15 important features based on mutual information [18], the embedding method selects non-zero coefficient features through LASSO regression [19,20], and the wrapping rule uses recursive feature elimination (RFE) combined with a random forest model to retain the top 15 important features [21–23]. The second stage integrates the three sets of candidate features through a hard voting mechanism, and determines the features with a screening frequency ≥ 2 as the final feature subset. This method innovatively combines the model independence of the filtering method, the regularization constraint of the embedding method, and the performance-oriented characteristics of the wrapping method. Through the majority voting mechanism, it effectively improves the stability and generalization ability of feature selection, and is particularly suitable for ensemble learning scenarios where the prediction performance of the base models is similar but there are differences [24].

This study uniformly uses support vector machines (SVMs) as classification models. The ability of its kernel function to process high-dimensional feature spaces effectively avoids the dimensionality curse problem and is particularly suitable for verifying feature subsets after filtering and embedding methods. Based on the principle of structural risk minimization, SVM gives the model natural robustness to feature redundancy and noise, and can objectively evaluate the changes in the generalization performance of the wrapping method during iterative feature elimination. In order to comprehensively evaluate the effect of feature selection, indicators such as the area under the ROC curve (AUC), accuracy (ACC), specificity (SPE), and sensitivity (SEN) [25] are used to systematically compare the performance of filtering, embedding, wrapping, and ensemble voting methods in feature selection.

## Construction of a screening model based on integrated soft voting

Based on the blood routine blood test data after feature selection, KNN [26–28], LR [29–31], RF [32–34], DT [35,36] and SVM [37–39] methods were used. Five machine learning algorithms were used to construct an AR early screening model. These five machine learning algorithms have the following characteristics for binary classification: KNN (K-Nearest Neighbor) calculates the distance between samples and uses majority voting among neighbors for classification. It is suitable for complex data distributions and sensitive to local patterns; LR (Logistic Regression) uses the sigmoid function to achieve probability mapping and has excellent interpretability; RF (Random Forest) uses bootstrap sampling to construct multiple decision trees and ensemble voting, effectively handling feature interactions; DT (Decision Tree) recursively partitions the feature space based on information gain, generating interpretable tree-like decision rules; SVM (Support Vector Machine) constructs an optimal hyperplane by maximizing the classification margin. Its kernel function can handle nonlinear problems and is robust to small sample sizes. These algorithms provide different solutions to classification problems from different perspectives. The AUC value is used as the evaluation indicator to compare the five machine learning algorithms, and the three methods with the highest AUC values are selected to build an integrated model.

This study used an 8:2 split ratio to divide the experimental data, assigning 1036 samples to the training set and 259 samples to the test set. During the splitting process, stratified sampling was used to maintain the positive/negative ratios in the training and test sets consistent with the original datasets to ensure a balanced data distribution. Finally, all feature data were normalized (Z-score) to eliminate the impact of dimensional differences between different detection metrics on the model. The Z-score standardization calculation formula is as follows (1):

$$Z = \frac{\S - \mu}{\sigma}$$

(1)

Where $\S$ represents the original data value, $\mu$ represents the mean of the data set, and $\sigma$ represents the standard deviation of the data set.

This study uses four evaluation indicators to comprehensively evaluate model performance: area under the receiver operating characteristic curve (AUC), accuracy, recall (also known as sensitivity), and specificity. The calculation formulas for each indicator are as follows (2–4):

$$\text{Accuracy} = \frac{\text{TP} + \text{TN}}{\text{TP} + \text{TN} + \text{FP} + \text{FN}} \tag{2}$$

$$\text{Recall} = \frac{\text{TP}}{\text{TP} + \text{TN}} \tag{3}$$

$$\text{Specificity} = \frac{\text{TN}}{\text{TN} + \text{FP}} \tag{4}$$

In the evaluation of medical diagnostic models, each indicator has clear clinical significance: AUC is a comprehensive indicator of the overall discriminative efficiency of the model, and its value range of 0.5–1.0 reflects the model's ability to distinguish between disease and non-disease states. The closer the value is to 1, the better the discriminative efficiency; Accuracy represents the overall proportion of correct predictions of all samples by the model; Recall specifically refers to the proportion of correctly identified samples in actual positive samples, reflecting the disease detection ability; Specificity indicates the proportion of correctly excluded samples in actual negative samples, reflecting the accuracy of disease exclusion. True positive (TP, actually positive and correctly predicted), true negative (TN, actually negative and correctly predicted), false positive (FP, actually negative but incorrectly predicted as positive) and false negative (FN, actually positive but incorrectly predicted as negative), these basic indicators constitute the confusion matrix for model performance evaluation.

The integrated model uses the soft voting method [40], taking the AUC value of each method as the weight and calculating the weighted average of the prediction probabilities of the three methods as the final prediction probability of AR, that is:

$$p = (AUC_1 * p_1 + AUC_2 * p_2 + AUC_3 * p_3)/3 \tag{5}$$

Where $p$ is the final predicted probability and where $AUC_i$ and $p_i$ are the AUC and predicted probability, respectively, of the different models. ROC curves and AUC values were used to evaluate the performance of five single models and integrated models, including KNN, LR, RF, DT and SVM, in the early screening of AR to verify the superiority of the integrated voting method. The overall experimental process is shown in Fig 1 below.

The core theoretical foundation of the SHAP (SHapley Additive exPlanations) method stems from game theory. Its core goal is to fairly distribute the model's prediction results to each input feature, thereby quantifying the contribution of each feature to the prediction. In a SHAP visualization chart, the Y-axis is arranged in descending order of feature importance, with the top feature representing the feature with the greatest impact on the model's overall output. Specifically, the X-axis of the summary bar chart represents the average SHAP absolute value of the feature; a larger value indicates a greater average change in the model's prediction results. The X-axis of the beeswarm plot represents the SHAP value, where a center position indicates that the feature has no impact on the prediction result, a rightward shift indicates a positive impact, and a leftward shift indicates a negative impact. The colors in the chart correspond to the actual value of the feature in the sample, with red representing a higher value and blue representing a lower value, providing a visual representation of the correlation between feature values and the direction of influence.

## Results

### Feature selection results

After multiple adjustments and optimizations, the final feature selection method and parameter settings are as follows: The filtering method (mutual information) directly calculates the correlation between feature data and labels without additional parameters, and selects the top 15 features as the screening results; the embedding method (LASSO) is built based

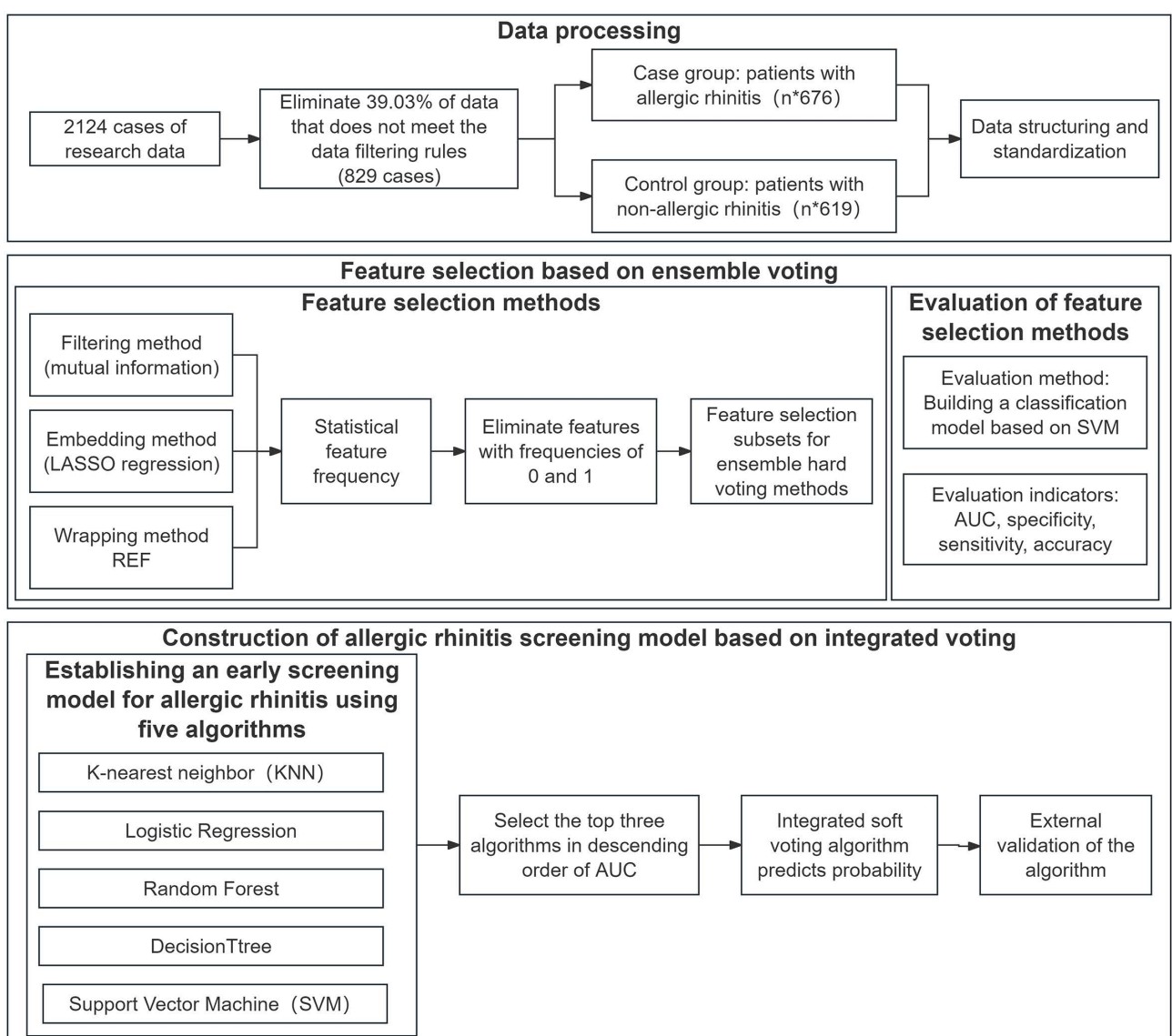

**Fig 1. Flowchart of the establishment of an AR early screening model via the integrated voting method.**

on the logistic regression model, with the parameter configuration of penalty = 'l1', C=0.1, olver='liblinear', and class_weight = 'balanced'. The screening rule is to select features with the absolute value of the feature coefficient greater than 0; the RFE method uses the random forest model as the basic model. After parameter optimization, the random forest model parameters are set to n_estimators = 10 and the class weight is 'balanced'; RFE's n_features_to_select = 15 (retaining 15 important features) and step = 1 (eliminating 1 feature at each iteration).

Table 2 lists the feature variables selected by the three feature selection methods and their contribution rankings (sorted by absolute value). The intersection of the features of the three methods was analyzed using a Venn diagram(-Fig 2), and a hard voting method was used to determine 16 core features as input features for subsequent models: age (Age), sex (Sex), EO#, EO%, HCT, HGB, MCHC, MCV, MONO%, NEUT%, PDW, P-LCR, PLT, RBC, RDW-CV, and RDW-SD.

**Table 2. Feature screening results based on filtering, wrapping, and embedding methods.**

| Feature selection method | Filtering (mutual information) | | Wrapping (REF) | | Embedding (LASSO) | |
|---|---|---|---|---|---|---|
| | Feature Name | Contribution value | Feature Name | Contribution value | Feature Name | Contribution value |
| 1 | EO# | 0.079 | RDW-SD | 0.102 | RDW-CV | 0.315 |
| 2 | RDW-SD | 0.063 | EO% | 0.101 | MPV | 0.132 |
| 3 | EO% | 0.052 | EO# | 0.097 | RDW-SD | −0.129 |
| 4 | age | 0.034 | age | 0.091 | sex | 0.115 |
| 5 | MCH | 0.029 | HCT | 0.081 | PDW | −0.080 |
| 6 | RBC | 0.027 | PDW | 0.060 | EO% | 0.069 |
| 7 | RDW-CV | 0.024 | MCHC | 0.056 | MCV | 0.052 |
| 8 | sex | 0.022 | MCV | 0.054 | MONO% | −0.008 |
| 9 | MCHC | 0.022 | PLT | 0.053 | P-LCR | −0.008 |
| 10 | WBC | 0.015 | NEUT% | 0.053 | age | −0.007 |
| 11 | HCT | 0.014 | LYMPH# | 0.052 | NEUT% | 0.004 |
| 12 | PDW | 0.013 | HGB | 0.051 | HGB | −0.003 |
| 13 | NEUT% | 0.012 | RDW-CV | 0.051 | PLT | −0.001 |
| 14 | MONO% | 0.011 | RBC | 0.050 | MCHC | 0.000 |
| 15 | P-LCR | 0.010 | NEUT# | 0.050 | | |

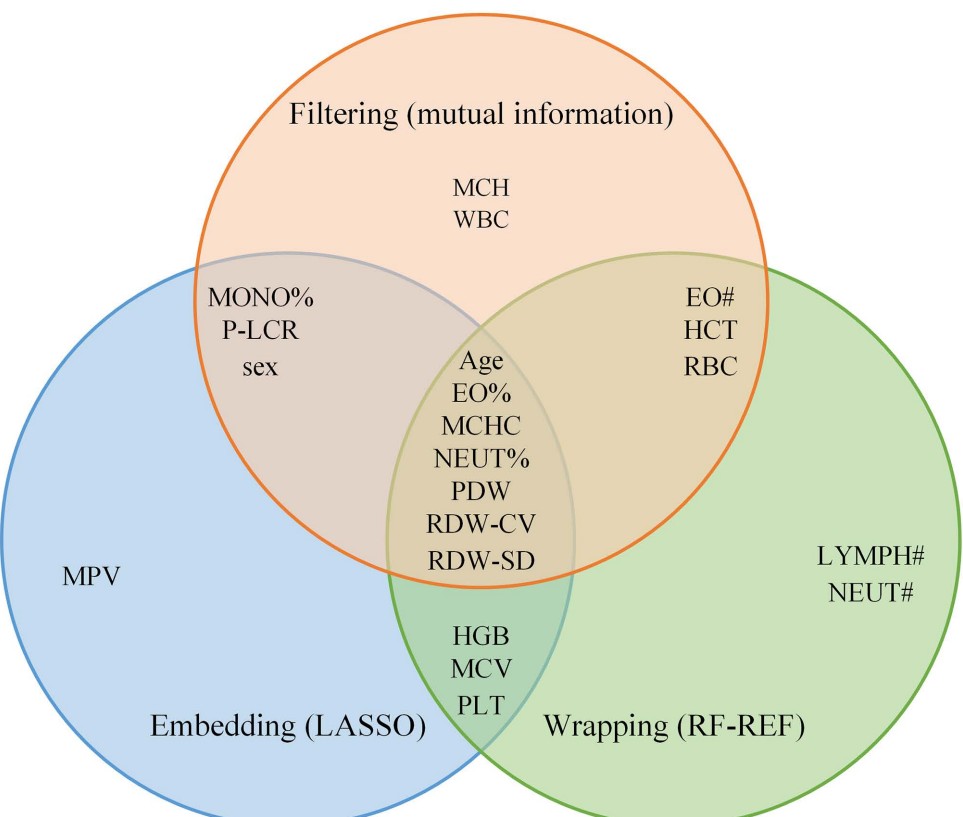

**Fig 2. Venn diagram of three sets of characteristic variables.**

To verify the impact of different feature selection methods on classification model performance, this study analyzed five feature sets: the initial 28-dimensional raw features, feature subsets selected using Mutual Information, RF-REF, and LASSO, and a set of 16 core features determined through hard voting. After constructing a classification model based on the support vector machine (SVM) algorithm, model performance was evaluated using the following metrics: AUC, Accuracy, Recall, and Specificity. The classification performance results for different feature sets on the SVM model are shown in Table 3. The analysis results show that the feature selection method based on ensemble hard voting combines the advantages of the three individual methods, resulting in the best classification model across all evaluation metrics (see Fig 3). Specifically, the core features selected using ensemble hard voting achieved the highest AUC (0.845), ACC (77.61%), Recall (75.56%), and Specificity (79.84%) in the SVM model. In summary, ensemble voting is a reasonable and effective feature selection method.

## Algorithm selection results

The model training process uses grid search technology to tune parameters and uses accuracy as the evaluation indicator to select the best parameter combination. Table 4 shows the model parameter settings, and the threshold parameters are all set to 0.5 by default.

**Table 3. Comparison of classification performance of SVM Models on different feature sets.**

| Evaluation Metrics / Feature Set | AUC | Accuracy | Recall | Specificity |
|---|---|---|---|---|
| Raw Features | 0.822 | 73.75% | 75.56% | 71.77% |
| Mutual Information | 0.835 | 75.68% | 74.81% | 76.61% |
| RF-REF | 0.825 | 74.13% | 75.56% | 72.58% |
| LASSO | 0.825 | 74.90% | 74.07% | 75.81% |
| Core Features | 0.845 | 77.61% | 75.56% | 79.84% |

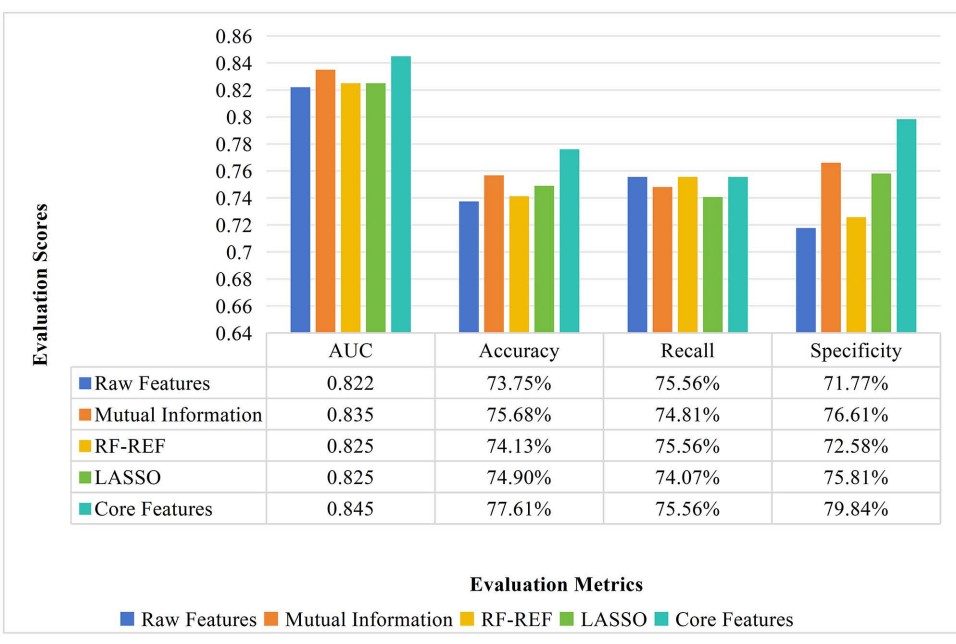

**Fig 3. Comparison of classification performance of SVM models on different feature sets.**

**Table 4. Model parameter setting.**

| Algorithm | Parameter | value |
|---|---|---|
| KNN | n_neighbors | 10 |
| | weights | uniform |
| | algorithm | auto |
| | p | 2 |
| LR | penalty | l2 |
| | C | 1 |
| | max_iter | 100 |
| | solver | sag |
| | random_state | 168 |
| DT | criterion | gini |
| | splitter | random |
| | max_features | auto |
| | max_depth | 15 |
| | random_state | 84 |
| RF | criterion | gini |
| | n_estimators | 90 |
| | max_features | auto |
| | random_state | 84 |
| SVM | kernel | rbf |
| | C | 1.4 |
| | gamma | 0.1 |
| | coef0 | 0 |
| | random_state | 42 |

We used the validation set to evaluate the performance of each model. The results revealed that the AUCs of the SVM, RF, and LR models were greater than 0.849, 0.836 and 0.827, respectively. To further improve the screening effect, the integrated soft voting method was used to calculate the final prediction probability based on the prediction probabilities of the SVM, RF, and LR models. The calculation formula is shown in Formula (6).

$$p = (0.827 * p_{LR} + 0.836 * p_{RF} + 0.849 * p_{SVM})/3 \tag{6}$$

where $p$ is the prediction probability of ensemble soft voting and where $p_{LR}$, $p_{RF}$, and $p_{SVM}$ are the prediction probabilities of the LR, RF, and SVM models, respectively. An example of an ensemble soft voting calculation is shown in Table 5.

The performance of each model was evaluated using a validation set. The evaluation metrics for the five machine learning methods under optimal parameters are shown in Table 6. The results show that the SVM, RF, and LR models achieved AUCs of 0.849, 0.836, and 0.827, respectively, and achieved the best performance in terms of specificity

**Table 5. Integrated soft voting calculation example.**

| Algorithm Example No. | LR | RF | SVM | Integrated soft voting |
|---|---|---|---|---|
| 1 | 90.87% | 92.22% | 97.88% | =(0.827*90.87%＋0.836*92.22%＋0.849*78.44%)/3＝78.44% |
| 2 | 94.59% | 97.78% | 96.73% | =(0.827*94.59%＋0.836*97.78%＋0.849*96.73%)/3＝80.69% |
| 3 | 34.30% | 40.00% | 50.78% | =(0.827*34.30%＋0.836*40.00%＋0.849*50.78%)/3＝34.97% |

**Table 6. Five machine learning evaluation metrics.**

| Machine Learning Models | AUC | Specificity | Accuracy | Recall |
|---|---|---|---|---|
| Ensemble Method | 0.862 | 0.855 | 0.772 | 0.696 |
| SVM | 0.849 | 0.815 | 0.788 | 0.763 |
| RF | 0.836 | 0.782 | 0.768 | 0.756 |
| LR | 0.827 | 0.742 | 0.768 | 0.793 |
| KNN | 0.771 | 0.734 | 0.718 | 0.704 |
| DT | 0.718 | 0.734 | 0.703 | 0.674 |

(0.815, 0.782, and 0.742), precision (0.788, 0.768, and 0.768), and recall (0.763, 0.756, and 0.793). To further improve performance, the prediction probabilities of SVM, RF, and LR were combined using an ensemble soft voting method to calculate the final result. Table 6 shows that the ensemble soft voting method achieved the highest AUC (0.862) and specificity (0.855). Its AUC and specificity were slightly higher than those of the SVM by 0.013 and 0.04, respectively. Only its accuracy (0.772) and recall (0.696) were slightly lower than those of the SVM, by 0.016 and 0.067, respectively. Overall, the ensemble soft voting method performed better. Fig 4 shows the ROC curves for different AR early screening methods, confirming this: the ensemble soft voting method achieved an AUC of 0.862, exceeding that of the other single models. Therefore, the ensemble soft voting method was ultimately chosen to combine the LR, RF, and SVM models to construct the AR screening model.

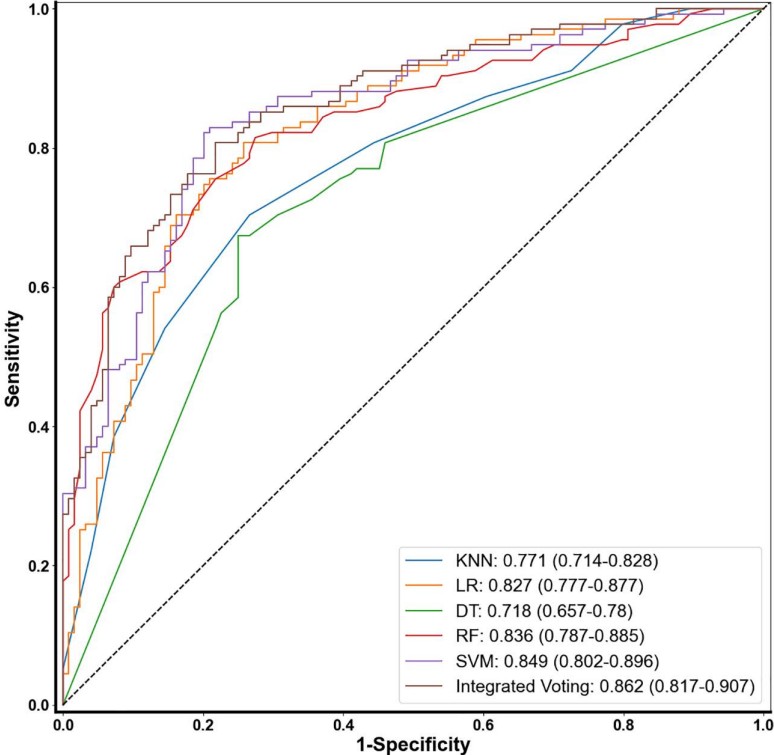

**Fig 4. Comparison of ROC curves for predicting AR via five machine learning algorithms and ensemble voting.**

Fig 5 shows the SHAP visualization results of the three best models, namely support vector machine (SVM), random forest (RF), and logistic regression (LR). The Support Vector Machine (SVM) model assigned the highest importance to RDW-SD, PDW, sex, P-LCR, and EO% (Fig 5A, 5B). RDW-SD (Red Blood Cell Distribution Width – Standard Deviation): This was identified as the most important feature in the SVM model. RDW-SD quantifies the heterogeneity in red blood cell size (anisocytosis). While not a traditional marker for AR, elevated RDW is increasingly recognized as a surrogate marker of systemic inflammation. The chronic inflammatory state in AR could potentially influence erythropoiesis, leading to greater variation in red cell size. In our analysis, higher RDW-SD values (red) correlated with increased SHAP values, suggesting a positive contribution to an AR prediction. PDW (Platelet Distribution Width) & P-LCR (Platelet Large Cell Ratio): These parameters measure the variability in platelet size. Larger, more reactive platelets are often associated with inflammatory and immune-mediated conditions. Platelets are known to participate in allergic inflammation by releasing mediators and interacting with other immune cells. The importance of PDW and P-LCR in our model suggests a potential link between platelet activation and AR. The SHAP dependence plots show that elevated levels of these markers (red) contribute to a higher probability of AR. The inclusion of sex as a key feature is supported by well-documented epidemiological evidence. The prevalence and severity of allergic diseases, including AR, often exhibit significant differences between males and females, likely due to a complex interplay of hormonal, genetic, and environmental factors. Our model has captured this effect, with the specific SHAP value for each patient indicating the direction and magnitude of sex's contribution to their individual prediction. The Random Forest (RF) model identified RDW-SD and EO# among its most influential features, followed by EO% (Fig 5C, 5D). The reaffirmation of eosinophil-related parameters (EO#, EO%) across all models underscores their critical and non-redundant role in AR screening. The concomitant importance of RDW-SD across all three models suggests a previously underappreciated link between erythrocyte indices and allergic inflammation that warrants further investigation; In the Logistic Regression (LR) model, the top contributing features were RDW-SD, RDW-CV, and PDW(Fig 5E, 5F). Elevated red cell distribution width (RDW) values, indicating heterogeneity in red blood cell size (anisocytosis), have been increasingly associated with systemic inflammatory states. In the context of AR, a chronic inflammatory condition, elevated RDW may serve as a surrogate marker for underlying inflammatory processes. Similarly, Mean Platelet Volume (MPV) is a marker of platelet activation, which can be modulated by allergic and inflammatory responses.

In summary, the feature contribution analysis confirms that our ensemble model successfully leverages features with strong biological and clinical foundations. The prominence of eosinophil counts (EO#, EO%) directly corroborates the pathophysiological mechanism of AR, thereby enhancing the clinical interpretability and credibility of our intelligent screening tool. The recurring importance of RDW-related features points to a potential novel hematological dimension in AR that merits future research.

### External validation

The performance evaluation results of the AR early screening model based on the ensemble voting method on the external validation set are shown in Table 7. Compared with the actual clinical diagnosis results, the model showed good screening effect, with an accuracy of 0.739, an AUC of 0.722, and a high specificity (0.829). It was reliable in excluding non-cases, with a recall rate of 0.603 and a medium sensitivity, which can meet the basic clinical needs of early screening. The confusion matrix of the external validation set is shown in Fig 6, which contains 184 samples. The model predicted 92 true negatives (TN), 19 false positives (FP), 29 false negatives (FN), and 44 true positives (TP). The total number of correctly predicted samples was 136, corresponding to an accuracy of 0.739.

### Discussion

In recent years, the prevalence of AR has increased annually, and 37% of allergic people gradually develop allergic asthma within 5 years or rapidly develop allergic asthma in extreme weather conditions (such as thunderstorms). Some

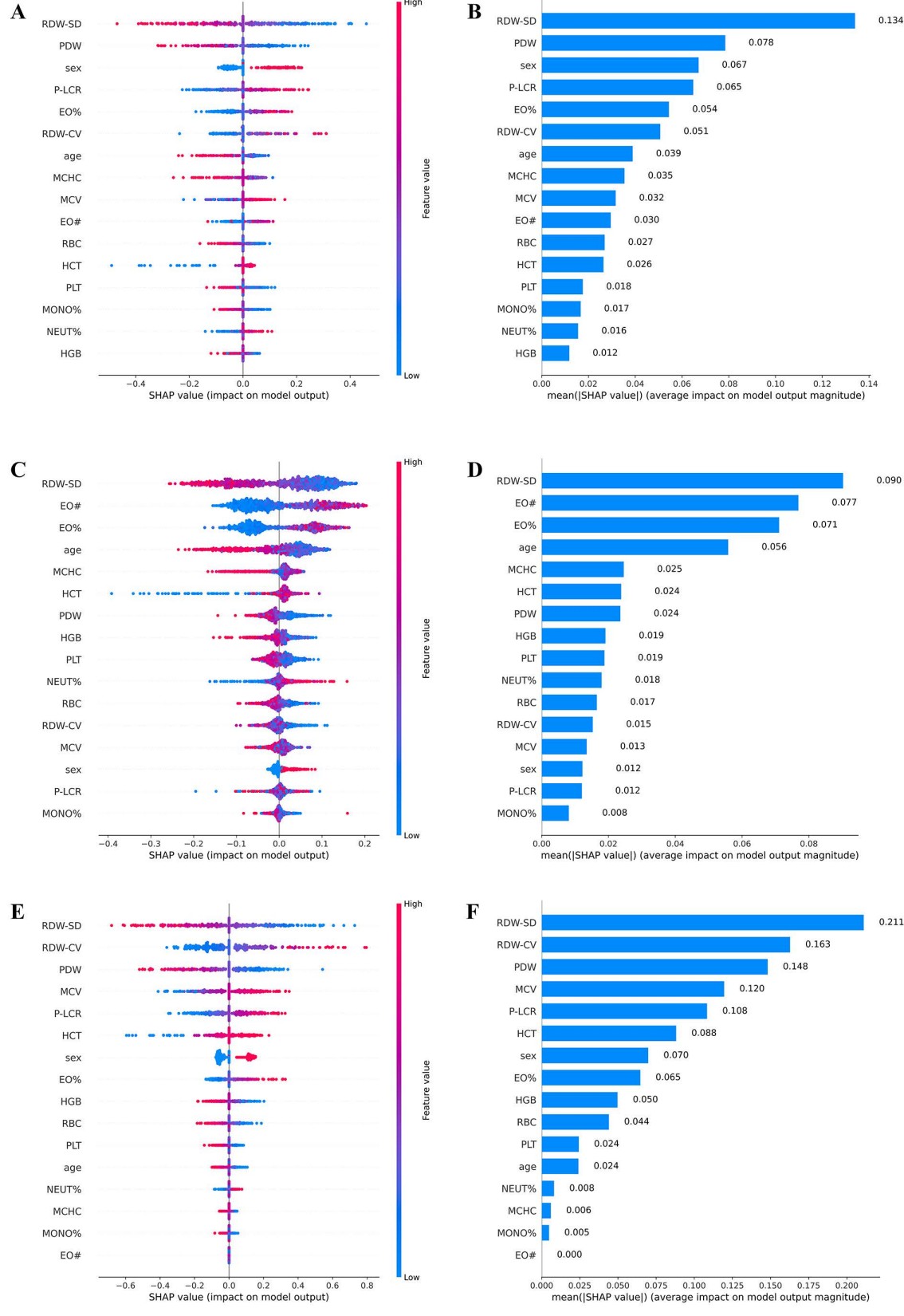

**Fig 5. SHAP interpretability analysis of the three best models: support vector machine (A, B), random forest (C, D), and logistic regression (E, F).**

**Table 7. Evaluation metrics of the external validation set.**

| Evaluation Metrics | Data Value | Evaluation Metrics | Data Value |
|---|---|---|---|
| AUC | 0.722 | TN | 92 |
| Accuracy | 0.739 | FP | 19 |
| Recall | 0.603 | FN | 29 |
| Specificity | 0.829 | TP | 44 |

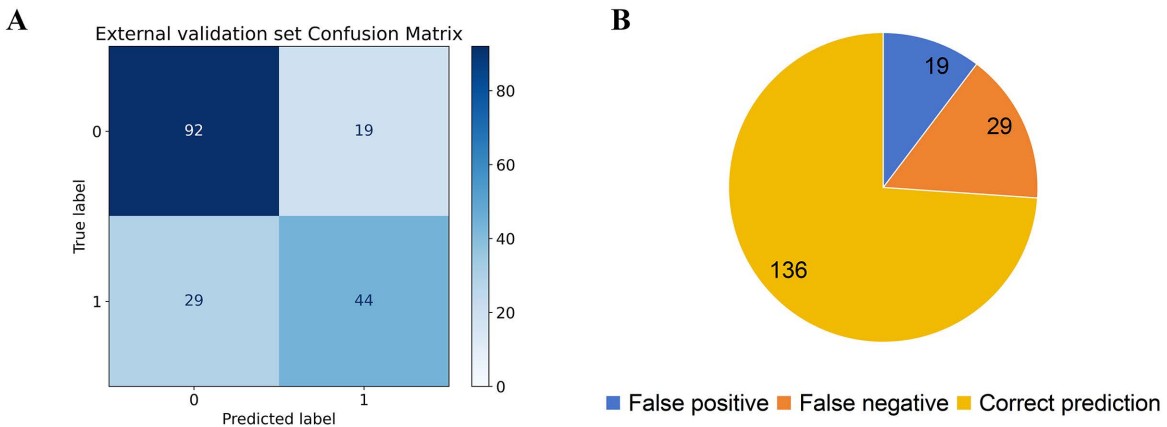

**Fig 6. External validation set performance evaluation results: (A) confusion matrix and (B) classification performance index analysis.**

patients may even experience anaphylactic shock due to the consumption of plant-derived foods related to allergic pollen [41]. The AR treatment guidelines propose the principle of "combining prevention and treatment, four-in-one" treatment; this includes secondary prevention, encompassing early detection, early diagnosis and early treatment to avoid or reduce the latent period during which allergic patients develop into AR patients [42]. Currently, the existing AR screening methods rely mainly on patients' symptom descriptions and allergen detection. However, the subjective descriptions of patients have a certain degree of ambiguity; allergen detection methods are limited in small- and medium-sized hospitals; and the cost of instruments, equipment, reagents and consumables is high, which increases the medical burden on patients. These challenges have led to slow progress in AR prevention in clinical practice. Therefore, it is crucial to establish a simple, effective, rapid and widely applicable AR early screening model, especially for AR auxiliary diagnosis in primary medical institutions, to standardize and guide clinicians in disease diagnosis and treatment more quickly.

Routine blood examination is a routine examination method in clinical medicine that can reveal the physiological and pathological status of the human body. Some studies have noted that routine blood and blood lipid biochemical indicators can assist doctors in diagnosing AR. The relationship between AR and blood cell components has gradually become a research hotspot. The literature [43] indicates that the blood eosinophil count and ratio may be good predictors of AR, especially AR accompanied by chronic sinusitis. The literature [44] indicates that the average platelet volume and platelet distribution width in children with AR are lower than those in healthy children, whereas the platelet/average platelet volume ratio is greater than that in healthy children. This study uses the integrated voting method to select 16 items from the original 28-dimensional features of the sample: RDW-SD, EO%, age, RDW-CV, MCHC, PDW, NEUT%, EO#, RBC, sex,

HCT, MONO%, P-LCR, MCV, HGB, and PLT. A classification model is built on the basis of the SVM algorithm to evaluate the impact of feature selection on the model. The performance from best to worst is as follows: integrated voting feature selection > single method feature selection > original features. In conclusion, the feature selection method proposed in this study not only significantly improves the accuracy and efficiency of AR screening but also opens a new path for the application of routine blood indicators in the diagnosis and treatment of allergic diseases. This discovery not only strengthens routine blood examination at the core position of AR diagnosis but also provides solid theoretical support and an empirical basis for clinical practice, encourages medical workers to explore the potential value of routine blood indices in early recognition of AR, condition monitoring and personalized treatment planning, and promotes refinement in the field of allergic rhinitis diagnosis and personalized development.

Machine learning technology can effectively process high-dimensional data and mine its inherent complex laws, so it has broad application prospects in the medical field. In recent years, researchers have begun to explore the application of machine learning technology for AR diagnosis. Christo et al. [45] used genetic algorithms and extreme learning machines to train features to achieve intelligent diagnosis of AR with an accuracy rate as high as 97.7%. Kavya et al. [46] used the random forest algorithm to construct a clinical decision support system for allergic diseases. The accuracy of the system was 86.39%, and the sensitivity for the combined rhinitis-urticaria category was 75%. However, existing methods have certain limitations in model construction, such as the use of a single algorithm for processing or the use of only symptoms and consultation information as inputs for analysis, resulting in subjective bias and instability in the prediction results. This study employed the integrated voting method to analyze the results of the KNN, LR, RF, DT, and SVM models and selected the top three AUC algorithms for integrated soft voting, aiming to obtain a more stable AR prediction probability. The results show that the AUCs of the integrated soft voting method are 3.5%, 2.6%, and 1.3% higher than those of the LR, RF, and SVM methods, respectively. In addition, this study used external verification data to verify the model, and the accuracy rate reached 73.91%. This finding shows that the integrated voting method can significantly improve the accuracy and stability of the model, providing a more accurate and reliable AR-assisted diagnostic tool for clinical use. In conclusion, this study, through the secondary development of nonspecific laboratory routine testing data, builds a model for grassroots medical institutions and early AR screening for grassroots medical service provision. Specifically, it opens a new opportunity, in the absence of advanced screening equipment and technical experts, for the performance of AR screening work without additional medical burdens, effectively filling the primary medical gaps in the field of allergic disease diagnosis and treatment. The successful implementation of this study provides a useful model for the early screening of similar diseases, which is expected to stimulate more medical innovations based on big data and further promote the overall upgrading and transformation of China's medical system.

Our model achieved an accuracy of 73.91% upon external validation, which is lower than the 97.7% accuracy reported by Christo et al. [45] using a genetic algorithm and extreme learning machine. This discrepancy can be primarily attributed to fundamental methodological differences. The study by Christo et al. likely utilized a feature set that included specific allergen test results or detailed symptom profiles, which are highly predictive but less universally available. In contrast, our model relies solely on routine blood test data, a more accessible and cost-effective but also more general and nonspecific data source. This strategic choice inherently trades off some predictive power for significantly greater applicability in primary care and resource-limited settings where advanced allergen testing is not feasible. Therefore, our model is not intended to replace specialized diagnostic tests but rather to serve as a widespread, first-line screening tool. Similarly, compared to the model by Kavya et al. [46] (accuracy: 86.39%), which incorporated clinical symptom data, our blood-based model offers the advantage of objectivity, minimizing the subjectivity associated with patient-self-reported symptoms.

A notable limitation of this study is that both the model development and external validation cohorts were exclusively recruited from a single center in Hohhot. This geographical specificity may introduce bias related to the local

population's genetic predispositions, predominant allergens (e.g., specific pollen types prevalent in the Inner Mongolia grassland region [6]), and regional environmental factors. Consequently, the generalizability of our screening model to populations in other geographical areas with different demographic characteristics and allergen exposures may be limited. While the internal and single-center external validation demonstrated promising performance, future validation on multi-center, geographically diverse cohorts is essential to confirm the robustness and widespread applicability of our model across various populations. In addition, this paper uses only routine blood test data when establishing the model and fails to integrate information such as contact history and symptoms of potential risk factors for AR, which reduces the accuracy and interpretability of the model. To better adapt to real clinical scenarios and improve the accuracy of decision-making, future work will aim to expand the data source and increase the information dimension of the model input.

This study utilized routine blood test data from a single time point, which does not account for the dynamic physiological changes inherent to allergic conditions. For instance, eosinophil counts (%)—a key feature in our model—are known to fluctuate with allergen exposure intensity, such as seasonal pollen variations [6,43]. A model trained on static measurements may therefore be less reliable for patients whose blood parameters are in a transient state. This limitation highlights a valuable direction for future research. Subsequent studies could implement longitudinal tracking of blood parameters in AR patients across different seasons. Developing a model that incorporates temporal patterns of change or that recommends diagnosis based on tests performed during symptomatic periods could significantly enhance predictive accuracy and clinical utility for seasonal AR sufferers.

## Supporting information

**S1 Table. Statistical results of routine blood test data of the external validation set based on SPSS.** All features of the external validation set (minimal dataset) were statistically analyzed using SPSS. Variables with a normal distribution (including near-normal distributions) were presented as mean ± standard deviation, while non-normally distributed variables were presented as quartiles. The statistical results are shown in Table 1. Using the ensemble voting method, this study ultimately identified 16 input features. Consequently, the statistical analysis of the external validation set included only these 16 features and was conducted using SPSS software.
(PDF)

## Acknowledgments

This study was supported by the Inner Mongolia Intelligent Big Data Research Institute team, whose valuable advice and support greatly contributed to the experimental guidance and manuscript revisions. We also extend our gratitude to colleagues from the Department of Allergy at Hohhot First Hospital for their assistance in data collection and analysis. Finally, we express our heartfelt thanks to our family and friends for their unwavering encouragement and understanding throughout this research endeavor.

## Author contributions

**Data curation:** Yanan Wang, Xin Tong, Shiyu Wu, Caiyan An, Huijiao Cai, Ruihuan Zhang.

**Formal analysis:** Biao Song.

**Methodology:** Yanan Wang, Ruihuan Zhang.

**Project administration:** Change Fan, Junjing Zhang, Biao Song.

**Writing – original draft:** Change Fan, Yanan Wang, Xin Tong.

**Writing – review & editing:** Change Fan, Yanan Wang.

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
