## [Decision Letter · Decision Letter 0]

22 Jul 2025

Dear Dr. Fan,

Specifically reviewers were concerned over the lack of detail in describing the parameter settings for the three feature selection methods. In addition, the rationale for selecting the top three base classifiers  lacked clarity. One reviewer also stated that the accuracy rate of 73.91% for a screening test requires additional clarification.

We look forward to receiving your revised manuscript.

Kind regards,

Colin Johnson, Ph.D.

Academic Editor

PLOS ONE

Journal Requirements:

Reviewers' comments:

Reviewer's Responses to Questions

**Comments to the Author**

1. Is the manuscript technically sound, and do the data support the conclusions?

Reviewer #1: Yes

Reviewer #2: Partly

2. Has the statistical analysis been performed appropriately and rigorously?

Reviewer #1: Yes

Reviewer #2: Yes

3. Have the authors made all data underlying the findings in their manuscript fully available?

Reviewer #1: Yes

Reviewer #2: Yes

4. Is the manuscript presented in an intelligible fashion and written in standard English?

Reviewer #1: Yes

Reviewer #2: No

Reviewer #1: Reviewer Comments:

I. Research Value and Innovation

Clear research significance with clinical application potential. The rising prevalence of Allergic Rhinitis (AR) and the limitations of current diagnostic methods (subjectivity, high cost, invasiveness) are well-established. This study builds an intelligent screening model using routine blood test data, enabling early screening without additional tests. This holds significant practical value for primary healthcare, aligning with trends in precision medicine and AI-assisted diagnosis.

Moderate methodological innovation. The ensemble voting approach integrating multiple feature selection techniques (filter, embedded, wrapper methods) with machine learning algorithms enhances feature selection reliability and model performance compared to single methods. The soft voting ensemble strategy, weighted by AUC to optimize classifier combination, is logically sound. Results demonstrate superior model performance over single algorithms (AUC=0.862, external validation accuracy 73.91%), providing methodological reference value.

II. Method Design and Implementation

Data Source and Preprocessing

Strengths: Data originates from a single center but has a substantial sample size (n=1295). Diagnostic criteria (combining symptoms and allergen testing) are clear. Grouping methods are reasonable.

Limitations:

Inclusion only of patients from Hohhot introduces geographical limitations potentially affecting model generalizability. This should be explicitly acknowledged in the Discussion.

Details of data preprocessing (e.g., missing value handling, outlier correction, feature standardization) are omitted, potentially impacting model stability. Supplementation is recommended.

Feature Selection and Model Construction

Strengths: The integrated hard voting method effectively cross-validates features across multiple strategies (retaining features with frequency ≥2). The final 16 retained features (e.g., eosinophil-related indices, RDW) align with AR immunopathological mechanisms (e.g., elevated eosinophils), demonstrating biological plausibility.

Limitations:

Specific parameter settings for the three feature selection methods (Mutual Information, LASSO, RF-RFE), such as the LASSO penalty coefficient or RF-RFE iteration count, are not detailed, hindering method reproducibility.

The rationale for selecting the top three base classifiers ("based on AUC value") lacks clarity regarding the threshold criterion. Supplementation with the AUC ranking results for all algorithms and statistical tests (e.g., Wilcoxon test) is recommended.

Model Evaluation

Strengths: Model generalizability is assessed using an external validation dataset. Multi-dimensional metrics (accuracy, AUC, etc.) are reported, and result visualization (confusion matrix) is clear.

Limitations:

The source, sample size, and feature distribution of the external validation set are not specified. Clarification is needed on whether it is an independent cohort or cross-center data.

Comparison with single algorithms is limited to AUC and accuracy. The absence of a calibration curve or Decision Curve Analysis (DCA) makes it difficult to comprehensively evaluate clinical net benefit.

III. Result Analysis and Discussion

Result Presentation

Strengths: The Venn diagram effectively illustrates the intersection of different feature selection methods. The model performance comparison figure (Fig 3) visually demonstrates the advantage of the ensemble method.

Limitations:

Feature importance ranking lacks discussion in the context of clinical significance (e.g., were eosinophil absolute count/percentage assigned the highest weights?). Supplementation with feature contribution analysis (e.g., SHAP values or Permutation Importance) is recommended.

Table 2 (Parameter Configuration) lacks complete presentation of some algorithm parameters (e.g., SVM kernel function, KNN neighbor count), hindering model replication.

The current figure legend for the confusion matrix (likely Fig 5) lists only "False positive," "False negative," and "Correct prediction" categories without specific numerical values or percentages. Adding this data would enhance clarity. Terminology must be consistent with definitions in the main text. Briefly define these terms in the legend or text. Color legends (■) must correspond precisely to chart colors. If figures are for potential B&W printing, use distinct patterns (e.g., hatching, dots) instead.

Regarding data presentation: The meaning of numbers like "136, 19, 29" below charts is unclear (possibly sample counts per category or other metrics). Explicit labeling is required to prevent misinterpretation. If sample counts, supplement with relevant statistical metrics (e.g., accuracy, recall) to bolster result credibility.

Formatting & Norms: Figure filenames (e.g., Fig5.tif) should follow journal style (e.g., "Figure5.tif" or "Fig5.png"). Ensure figure resolution meets journal requirements (typically ≥300 dpi).

Discussion Depth

Strengths: The Discussion objectively identifies study limitations (single data source, lack of symptom/exposure history integration) and proposes future directions (expanding data dimensions, incorporating multimodal information), presenting a clear logic.

Limitations:

Insufficient comparison with similar studies (e.g., Christo et al.'s 97.7% accuracy model). Analyze methodological differences (e.g., data type, algorithm choice) to clarify this model's specific applicability.

The impact of dynamic changes in blood routine indicators (e.g., seasonal fluctuations in seasonal AR patients) on the model is not explored and could be a valuable extension.

IV. Ethics and Writing Standards

Ethical Compliance. Mention of Institutional Review Board (IRB) approval is noted, but details regarding patient informed consent and data anonymization procedures are missing. Supplementation of the ethics statement details is required.

Writing and Formatting

Strengths: Well-structured manuscript. Abstract is concise. References cover recent relevant studies in ML and AR diagnosis.

Limitations:

Some figure/table labels are unclear (e.g., missing y-axis unit in Fig 3, inconsistent table numbering). Standardize figure/table formatting.

Redundant descriptions exist between the English abstract and main text (e.g., repeated steps of the ensemble voting in the Methods). Streamlining is advised.

V. Overall Recommendations

Major Revisions Required:

Data & Methods:

Supplement data preprocessing details (missing value handling, standardization methods) and feature selection parameter settings.

Clarify the source, sample size, and feature distribution of the external validation set, OR consider adding multi-center validation.

Supplement AUC rankings for all base classifiers and provide statistical test results to justify selecting the top three.

Authors must supplement charts/tables with specific data and explanations to ensure information is complete and easily understandable.

Check terminology and formatting for consistency with the journal's author guidelines.

Results & Discussion:

Add feature importance analysis (e.g., SHAP values) and interpret key features (e.g., EO#, RDW-SD) in the context of clinical literature and their biological significance.

Supplement calibration curve and decision curve analysis (DCA) to evaluate clinical utility.

Compare the advantages/disadvantages and innovation of this model against similar ML models for AR.

Provide deeper discussion of chart/table results, explicitly linking them to the research objectives.

Ethics & Formatting:

Enhance the ethics statement to include informed consent procedures and data privacy protection measures.

Correct figure/table labeling errors, unify table numbering, and eliminate redundant text.

Minor Revisions Suggested (Optional Optimizations):

Explore the interaction effect of demographic features (age, sex) with blood routine indicators on AR.

Compare other ensemble strategies (e.g., Stacking) with the soft voting method used here to further validate model robustness.

Summary: This study constructs an intelligent AR screening model based on routine blood test data. The methodology is sound, and the results hold clinical relevance. Enhancing data diversity, feature interpretability, and model evaluation dimensions would significantly strengthen its scientific rigor and persuasiveness. The authors are recommended to undertake a major revision addressing the points above before resubmission.

Reviewer #2: I agree that there are few objective tests for allergic rhinitis and that this poses several challenges.

I have a few comments

1. I have reservations about whether the blood tests presented in the study can be fully conducted at primary care facilities rather than tertiary hospitals. Screening tests for allergic rhinitis should ideally be low-cost and accessible at the primary care level.

2. The use of hard and soft voting methods combining KNN, LR, RF, DT, and SVM models appears to yield improved results compared to previous approaches. However, the accuracy rate of 73.91% for a screening test requires cautious interpretation. It is recommended that additional clarification or discussion of the limitations be provided to contextualize this result appropriately.

3. A variety of inference models are employed, and the selection process for these models is considered important. Although numerous references are cited, it would be beneficial to include a clear rationale for the selection of KNN, LR, RF, DT, and SVM. While the performance evaluation and validation process using voting methods is illustrated schematically, the presented models appear to be basic validation models. Therefore, it would be helpful to provide an explanation addressing the absence of more recent models to enhance the study's relevance and rigor.

4. While utilizing the abundant data available in China is advantageous, considering that academic journals are read by a global audience, it would be more beneficial to frame the study from a perspective that offers utility to all patients, rather than focusing solely on advancements in the Chinese healthcare system.

5. It would be nice to add demographic data

6. It might be good to have your English proofread. Overall, it is understandable, but a review of spacing and similar details is necessary.

**Do you want your identity to be public for this peer review?** For information about this choice, including consent withdrawal, please see our Privacy Policy

Reviewer #1: No

Reviewer #2: **Yes: ** Young Joon, Jun, M.D., Ph.D.

---

## [Author Response · Author response to Decision Letter 1]

21 Oct 2025

5. Review Comments to the Author

Reviewer #1: Reviewer Comments:

I. Research Value and Innovation

Clear research significance with clinical application potential. The rising prevalence of Allergic Rhinitis (AR) and the limitations of current diagnostic methods (subjectivity, high cost, invasiveness) are well-established. This study builds an intelligent screening model using routine blood test data, enabling early screening without additional tests. This holds significant practical value for primary healthcare, aligning with trends in precision medicine and AI-assisted diagnosis.

Moderate methodological innovation. The ensemble voting approach integrating multiple feature selection techniques (filter, embedded, wrapper methods) with machine learning algorithms enhances feature selection reliability and model performance compared to single methods. The soft voting ensemble strategy, weighted by AUC to optimize classifier combination, is logically sound. Results demonstrate superior model performance over single algorithms (AUC=0.862, external validation accuracy 73.91%), providing methodological reference value.

II. Method Design and Implementation

Data Source and Preprocessing

Strengths: Data originates from a single center but has a substantial sample size (n=1295). Diagnostic criteria (combining symptoms and allergen testing) are clear. Grouping methods are reasonable.

1�Inclusion only of patients from Hohhot introduces geographical limitations potentially affecting model generalizability. This should be explicitly acknowledged in the Discussion.

Response:

We thank the reviewer for the valuable comments. As requested, we have supplemented the relevant content in the Discussion section�A notable limitation of this study is that both the model development and external validation cohorts were exclusively recruited from a single center in Hohhot. This geographical specificity may introduce bias related to the local population's genetic predispositions, predominant allergens (e.g., specific pollen types prevalent in the Inner Mongolia grassland region [6]), and regional environmental factors. Consequently, the generalizability of our screening model to populations in other geographical areas with different demographic characteristics and allergen exposures may be limited. While the internal and single-center external validation demonstrated promising performance, future validation on multi-center, geographically diverse cohorts is essential to confirm the robustness and widespread applicability of our model across various populations.In addition, this paper uses only routine blood test data when establishing the model and fails to integrate information such as contact history and symptoms of potential risk factors for AR, which reduces the accuracy and interpretability of the model. To better adapt to real clinical scenarios and improve the accuracy of decision-making, future work will aim to expand the data source and increase the information dimension of the model input.

2�Details of data preprocessing (e.g., missing value handling, outlier correction, feature standardization) are omitted, potentially impacting model stability. Supplementation is recommended..

Response:

Thank you for your valuable feedback! We have supplemented and improved the data preprocessing process (including missing value handling, outlier handling, and feature standardization). The specific additions are as follows: the method for handling missing and outlier features has been added in line 127; the specific method and related formula for data standardization have been added in line 232.

Modify the content in line 127�“Eliminate sample data with missing values or outliers. Finally, routine examination data and AR diagnosis results were collected for 1295 subjects, including 676 AR patients and 619 non-AR patients.”

Modify the content in line 232�“ Finally, all feature data were normalized (Z-score) to eliminate the impact of dimensional differences between different detection metrics on the model. The Z-score standardization calculation formula is as follows (1):

(1)”

3�Specific parameter settings for the three feature selection methods (Mutual Information, LASSO, RF-RFE), such as the LASSO penalty coefficient or RF-RFE iteration count, are not detailed, hindering method reproducibility.

Response:

Thank you for your valuable feedback! We indeed neglected to explain the specific parameter settings for the three feature selection methods. We have now supplemented this section in the paper, located in the "Feature Selection Results" section on line 291. Specifically, the mutual information method performs feature selection by directly calculating the correlation between the data and the label, requiring no additional parameter settings. The LASSO method is based on logistic regression, and its parameters are the same as those of the logistic regression method. The RFE method is based on the random forest method, and its corresponding parameters are also the parameters of the random forest method.

Modify the content in line 291�“After multiple adjustments and optimizations, the final feature selection method and parameter settings are as follows: The filtering method (mutual information) directly calculates the correlation between feature data and labels without additional parameters, and selects the top 15 features as the screening results; the embedding method (LASSO) is built based on the logistic regression model, with the parameter configuration of penalty='l1', C=0.1, olver='liblinear', and class_weight='balanced'. The screening rule is to select features with the absolute value of the feature coefficient greater than 0; the RFE method uses the random forest model as the basic model. After parameter optimization, the random forest model parameters are set to n_estimators=10 and the class weight is 'balanced'; RFE's n_features_to_select=15 (retaining 15 important features) and step=1 (eliminating 1 feature at each iteration).”

4�The rationale for selecting the top three base classifiers ("based on AUC value") lacks clarity regarding the threshold criterion. Supplementation with the AUC ranking results for all algorithms and statistical tests (e.g., Wilcoxon test) is recommended.

Response:

Thank you very much for your valuable suggestions! After careful verification, we did have the problem of insufficient explanation of the basis for selecting the three basic classifiers, and lacked clear explanation in the presentation of the results of the three feature selection methods. Based on this, we have made targeted supplements and improvements in the paper: the selection logic and theoretical support of the three basic classifiers are explained in detail on line 162; the model experimental results of the three feature selection methods are supplemented on line 331 to verify their effectiveness with objective data; the specific analysis of the feature contribution value is presented on line 311 to quantify the influence of each feature on the model performance; at the same time, the statistical test results of the input features are supplemented on line 158, and the test methods used are explained in detail on line 144 to ensure the transparency and reproducibility of the research methods. The above modifications are intended to improve the rigor and completeness of the paper. Thank you again for your professional guidance!

Modify the content in line 162�“In terms of feature selection, filtering, embedding and wrapping methods each have their own advantages: filtering uses statistical indicators (such as chi-square test and mutual information) to perform preliminary feature screening independently of the model, avoiding model assumption bias and being suitable for high-dimensional data preprocessing [14]; embedding embeds feature selection into the model training process (such as LASSO or feature importance of tree models), automatically eliminates redundant features through regularization or integration mechanisms, and is particularly suitable for processing nonlinear relationship features [15,16]; wrapping is guided by model performance (such as recursive feature elimination, RFE), and captures feature interaction effects through iterative search, and performs outstandingly in scenarios such as medical diagnosis that require fine optimization of feature combinations [17]. Based on the above advantages, this study selected these three methods for feature screening experiments.”

Modify the content in line 331�“Table 3. Comparison of Classification Performance of SVM Models on Different Feature Sets

Evaluation

Metrics

Feature Set AUC Accuracy Recall Specificity

Raw Features 0.822 73.75% 75.56% 71.77%

Mutual Information 0.835 75.68% 74.81% 76.61%

RF-REF 0.825 74.13% 75.56% 72.58%

LASSO 0.825 74.90% 74.07% 75.81%

Core Features 0.845 77.61% 75.56% 79.84%

”

Modify the content in line 311�“Table 2. Feature screening results based on filtering, wrapping, and embedding methods

Feature selection method Filtering (mutual information) Wrapping (REF) Embedding (LASSO)

Feature Name Contribution value Feature Name Contribution value Feature Name Contribution value

1 EO# 0.079 RDW-SD 0.102 RDW-CV 0.315

2 RDW-SD 0.063 EO% 0.101 MPV 0.132

3 EO% 0.052 EO# 0.097 RDW-SD -0.129

4 age 0.034 age 0.091 sex 0.115

5 MCH 0.029 HCT 0.081 PDW -0.080

6 RBC 0.027 PDW 0.060 EO% 0.069

7 RDW-CV 0.024 MCHC 0.056 MCV 0.052

8 sex 0.022 MCV 0.054 MONO% -0.008

9 MCHC 0.022 PLT 0.053 P-LCR -0.008

10 WBC 0.015 NEUT% 0.053 age -0.007

11 HCT 0.014 LYMPH# 0.052 NEUT% 0.004

12 PDW 0.013 HGB 0.051 HGB -0.003

13 NEUT% 0.012 RDW-CV 0.051 PLT -0.001

14 MONO% 0.011 RBC 0.050 MCHC 0.000

15 P-LCR 0.010 NEUT# 0.050

”

Modify the content in line 144�“Before model construction, to optimize the effect of feature screening, this study used SPSS software for statistical analysis based on 28-dimensional features (26 routine blood test indicators and age and gender). The statistical methods selected were as follows: (1) Qualitative variables were tested using the chi-square test and expressed as categories (percentages); (2) Quantitative variables were first tested for normality. If the variables were normal (Shapiro-Wilk test p>0.05) or approximately normal distribution (standard deviation < mean/3), the independent sample t-test (mean ± standard deviation) was used, and significance was determined by the Levene test (Student's t-test was used when p>0.05, otherwise the Welch's t-test was used); (3) Non-normal or approximately normal distribution variables were tested using the Mann-Whitney U test and expressed as median (upper quartile, lower quartile). Specific feature abbreviations, definitions, and statistical results are shown in Table 1.”

Modify the content in line 158�“Table 1. Statistical results of blood routine test data based on SPSS.

Characteristic Full name Negative and positive statistical results Positive result Negative result P-value

sex 0(61.69%) 0(75.61%) 0(68.34%) <0.001

age 43.38±11.74 40.65±10.17 46.36±12.59 <0.001

RBC Red blood cell count 4.8±0.46 4.8±0.47 4.81±0.45 0.687

MCV Mean corpuscular volume 92.02±5.46 91.84±4.82 92.21±6.09 0.224

PDW Platelet distribution width 13.51±1.8 13.3±1.71 13.73±1.87 <0.001

WBC White blood cell count 6.34±1.6 6.37±1.57 6.32±1.64 0.528

NEUT% Neutrophil ratio 56.69±7.79 56.77±7.73 56.61±7.87 0.706

LYMPH% Lymphocyte ratio 33.99±7.42 33.35±7.22 34.69±7.58 0.001

EO% Eosinophil ratio 1.7(1.,3.) 2.4(1.4,3.9) 1.3(0.8,2.) <0.001

BASO% Basophil ratio 0.5(0.3,0.6) 0.5(0.3,0.6) 0.5(0.3,0.6) 0.459

NEUT# Neutrophil absolute value 3.49(2.8,4.26) 3.66±1.21 3.49(2.76,4.26) 0.385

BASO# Absolute basophil count 0.03(0.02,0.04) 0.03(0.02,0.04) 0.03(0.02,0.04) 0.408

HGB Hemoglobin 143.23±17.02 143.03±16.46 143.44±17.62 0.671

HCT Hematocrit 0.42±0.11 0.44±0.04 0.43(0.4,0.47) <0.001

MCH Mean corpuscular hemoglobin content 29.81±2.15 29.81±1.9 29.82±2.39 0.912

MCHC Mean corpuscular hemoglobin concentration 323.88±11.42 324.49±10.49 323.22±12.33 0.047

RDW-CV Red blood cell distribution width 13.37±1.48 13.19±1.41 13.57±1.53 <0.001

PLT Platelet count 256.9±60.81 254.34±55.5 259.69±66.06 0.117

MPV Mean platelet volume 11.13±1.21 11.11±1.23 11.15±1.2 0.605

PCT Platelet count 0.28±0.06 0.28±0.06 0.29±0.07 0.099

MONO# Absolute value of monocytes 0.4±0.13 0.4±0.13 0.41±0.14 0.634

MONO% Monocyte ratio 6.42±1.5 6.38±1.47 6.48±1.53 0.219

EO# Absolute eosinophil count 0.11(0.06,0.18) 0.15(0.08,0.23) 0.08(0.05,0.13) <0.001

IG# Immature granulocyte count 0.01(0.01,0.01) 0.01(0.01,0.01) 0.01(0.01,0.01) 0.484

RDW-SD Red blood cell distribution width - SD 44.16±3.57 43.22±3.48 45.18±3.39 <0.001

IG% The percentage of immature granulocytes 0.1(0.1,0.2) 0.1(0.1,0.2) 0.1(0.1,0.2) 0.843

P-LCR Large platelet ratio 30.78±7.85 30.37±7.56 31.23±8.14 0.049

LYMPH# Absolute lymphocyte count 2.12±0.57 2.08±0.54 2.15±0.6 0.034

”

5�The source, sample size, and feature distribution of the external validation set are not specified. Clarification is needed on whether it is an independent cohort or cross-center data.

Response:

Thank you for the valuable comments provided by the reviewers. We have supplemented the detailed information of the external validation set as requested and made the corresponding revisions in the main text.

Modify the content of line 131�“This study used a single-center prospectively recruited external validation cohort. The data came from outpatients in the Allergy Department of Hohhot First Hospital from March 21 to September 3, 2023. After screening and meeting the inclusion criteria, a total of 184 patients aged 19-70 years (49 males and 135 females) were included; according to clinical gold standards (such as allergen sIgE testing and typical symptoms and signs), they were divided into 73 allergic rhinitis (AR) patients and 111 non-AR control groups.”

6�Comparison with single algorithms is limited to AUC and accuracy. The absence of a calibration curve or Decision Curve Analysis (DCA) makes it difficult to comprehensively evaluate clinical net benefit.

Response:

Thank you for your suggestions. Regarding model calibration analysis, inherent limitations of the dataset currently present significant challenges. Although the models generalize well in training and testing, the ensemble models (SVM, LR, and RF) struggle to correct for probability bias due to inappropriate calibration methods. Furthermore, the boundary between AR and healthy individuals is blurred (e.g., it's difficult to distinguish between mild cases and those with allergies). Some "gray zone" samples lead to fluctuations in predictions within the intermediate probability range (0.4–0.6), causing the calibration curve to deviate from the ideal line. Furthermore, the limited amount of available calibration data results in low stability and a lack of true reflection of test set performance. Therefore, the current calibration plots are ineffective. We plan to reanalyze the data distribution through data augmentation, SMOTE, or an external validation set to improve the data distribution. Furthermore, this study pre-determined a classification threshold of 0.5 to correspond to clinical treatment decisions (positive, recommend treatment; negative, no recommendation or further testing). Because AR treatment is low-risk, reversible, and has mild adverse reactions, the clinical significance of assessing the benefit-risk trade-off at different thresholds is limited, so a decision curve analysis (DCA) was not performed.

7�Feature importance ranking lacks discussion in the context of clinical significance (e.g., were eosinophil absolute count/percentage assigned the highest weights?). Supplementation with feature contribution analysis (e.g., SHAP values or Permutation Importance) is recommended.

Response:

Sincerely thank you for your valuable suggestions. Given that this experiment utilizes an ensemble model cons

---

## [Decision Letter · Decision Letter 1]

11 Nov 2025

Construction of an Intelligent Screening Model for Allergic Rhinitis Based on Routine Blood Tests

PONE-D-25-12740R1

Dear Dr. Fan,

We’re pleased to inform you that your manuscript has been judged scientifically suitable for publication and will be formally accepted for publication once it meets all outstanding technical requirements.

Kind regards,

Colin Johnson, Ph.D.

Academic Editor

PLOS ONE

Additional Editor Comments (optional):

Reviewers' comments:

Reviewer's Responses to Questions

**Comments to the Author**

Reviewer #1: All comments have been addressed

Reviewer #2: All comments have been addressed

2. Is the manuscript technically sound, and do the data support the conclusions?

Reviewer #1: Yes

Reviewer #2: Yes

3. Has the statistical analysis been performed appropriately and rigorously?

Reviewer #1: Yes

Reviewer #2: Yes

4. Have the authors made all data underlying the findings in their manuscript fully available?

Reviewer #1: Yes

Reviewer #2: Yes

5. Is the manuscript presented in an intelligible fashion and written in standard English?

Reviewer #1: Yes

Reviewer #2: Yes

Reviewer #1: The author has refined the revision suggestions, made thorough revisions and agreed to accept this research article.

Reviewer #2: The previously mentioned points have been appropriately addressed.

As a screening tool, it may have clinical utility; however, broader, generalizable use will require further research.

The authors appear to have recognized this point and acknowledged it as a limitation.

**Do you want your identity to be public for this peer review?** For information about this choice, including consent withdrawal, please see our Privacy Policy

Reviewer #1: No

Reviewer #2: **Yes: ** Young Joon Jun

---

## [Editor Report · Acceptance letter]

PONE-D-25-12740R1

PLOS ONE

Dear Dr. Fan,

I'm pleased to inform you that your manuscript has been deemed suitable for publication in PLOS ONE. Congratulations! Your manuscript is now being handed over to our production team.

Kind regards,

on behalf of

Dr. Colin Johnson

Academic Editor

PLOS ONE